# The Association between Mobile Phone Use and Severe Traffic Injuries: A Case-Control Study from Saudi Arabia

**DOI:** 10.3390/ijerph16152706

**Published:** 2019-07-29

**Authors:** Suliman Alghnam, Jawaher Towhari, Mohamed Alkelya, Ahmad Alsaif, Mohamed Alrowaily, Fawaz Alrabeeah, Ibrahim Albabtain

**Affiliations:** 1Population Health Section-King Abdullah International Medical Research Centre (KAIMRC), King Saud Bin Abdulaziz University for Health Sciences (KSAU-HS), Riyadh 11426, Saudi Arabia; 2College of Medicine, King Saud bin Abdulaziz University for Health Sciences, Riyadh 11426, Saudi Arabia; 3Health Research Department, Saudi Health Council, Riyadh, 13315, Saudi Arabia; 4Community Medicine Department, King Khalid University Hospital, Riyadh 12372, Saudi Arabia; 5King Abdulaziz Medical City, National Guard Healthcare Affairs, Riyadh 11426, Saudi Arabia; 6Department of Surgery, King Abdulaziz Medical City, National Guard Healthcare Affairs, Riyadh 11426, Saudi Arabia

**Keywords:** road injuries, mobile phone, seatbelt, HRQOL, Saudi Arabia

## Abstract

Road traffic injury (RTI) is the third leading cause of death in Saudi Arabia. Using a mobile phone when driving is associated with distracted driving, which may result in RTIs. Because of limited empirical data, we investigated the association between mobile phone use and RTI in injured patients and community controls in Riyadh. Cases were patients admitted to King Abdulaziz Medical City (KAMC) between October 2016 and March 2018 due to RTIs. During admission, mobile phone use at the time of the accident was investigated. The controls were drivers observed at various locations citywide. A logistic regression model was constructed to estimate the association between mobile phone use while driving and sustaining RTIs. We included 318 cases and 1700 controls. For the cases, using a mobile phone was associated with higher severity and prevalence of disability. In addition, using a mobile phone while driving is associated with 44% higher odds of incurring a severe RTI (*p* = 0.04). Mobile phone use while driving is prevalent in Riyadh and pose a significant threat of disability. In addition, the low prevalence of seatbelt use is alarming and requires significant improvement. Prevention programs may use these findings to educate the public and policymakers and to advocate for increased visibility of enforcement to reduce RTIs and improve population health.

## 1. Introduction

Traffic accidents are a significant threat to public health in every country. Globally, as many as 50 million individuals sustain a road traffic injury (RTI) every year [1]. From a financial perspective, the estimated global cost of RTIs is approximately $518 billion, of which more than $65 billion is spent in developing countries [2]. 

In Saudi Arabia (SA), RTI is the third leading cause of death and the leading cause of years of life lost (YLLs) [3]. According to the World Health Organization, SA has a traffic mortality rate of 24.8 per 100,000, which is substantially higher than other countries (i.e., Qatar: 14 per 100,000) with a similar demographic profile and driving environment [4]. Notably, RTIs account for over 80% of all trauma admissions, and at any moment, 20% of hospital beds are occupied by RTI patients [5,6]. While the burden of RTIs is decreasing in developed countries, it is increasing in developing countries such as SA. In 2016, there was an 8% increase in mortality compared to the preceding year [7]. Evidently, traffic accidents pose a significant burden on population health. 

Several factors contribute to the incidence of RTIs [8,9]. These may include suboptimal road conditions, unsafe motor vehicles, and risky driving behaviors [10]. Speeding, running a red light, driving under the influence, seatbelt nonuse, and distracted driving are examples of common traffic violations associated with RTIs globally and in SA [11]. Research quantifying the contribution of these risk factors may facilitate planning preventative strategies.

Distracted driving is an emerging threat to traffic safety [12]. There are many forms of distraction, including manual, visual, auditory, or cognitive [13]. For example, drivers can be distracted visually by looking away from the road, or physically by holding or using devices such as mobile phones [14]. Globally, there is an increased risk of RTIs due to distracted driving associated with mobile phone use [15,16,17]. The use of mobile phones while driving has been linked to a slower reaction time, less control, and less attention to relevant visual information [18]. The increased use of smartphones and engagement in social media may also have contributed to increasing distracted driving and, consequently, the incidence and severity of RTIs [19]. 

Literature from developed countries report that using a mobile phone while driving is associated with an increased risk of traffic accidents. However, these results may not necessarily apply to the Saudi population [20]. The prevalence of mobile phone use while driving may vary from one country to another due to differences in driving environments or behaviors. Literature suggests that aggressive and risk-taking behavior is prevalent among Saudi drivers [21]. 

There is a paucity of empirical data about mobile phone use while driving in SA. Previous local studies were based on self-report surveys among the general population focusing on habitual use [22,23,24]. Concurrent with increased mobile phone subscriptions in recent years, a recent study found an increase in RTI severity beginning in 2014, possibly due to distracted driving [25]. There is a need to fully understand the impact of mobile phone use while driving on traffic safety in SA to guide policy development and facilitate effective prevention. 

Though it is known that thousands of individuals die annually in SA due to RTIs, the impact on survivors is under-researched. Specifically, to what extent does RTIs affect health-related quality of life (HRQOL) of individuals who survived an RTI? In addition, to our best knowledge, no previous studies investigated the association between mobile phone-related RTIs and health outcomes. Police traffic accident reports do not include information about mobile phone use or other relevant factors such as seatbelt use that may play a role in RTIs and their outcomes. In addition, the information may not have been captured accurately. A study reported that police underestimate annual deaths due to traffic accidents [26]. Obtaining information about mobile phone and seatbelt use from survivors could be a starting point to evaluate their impact on RTIs. 

Evaluating the HRQOL of injured patients may provide a basis to determine the burden of non-fatal injuries in SA. The results could also offer an opportunity to compare decrements following an RTI with other countries, as well as an evidence-base for interventions to improve the health of those affected. Therefore, our study aims to (1) evaluate the association between mobile phone use and RTI, and (2) assess the HRQOL of individuals who sustained an RTI. We hypothesize that mobile phone use is associated with an increased risk and severity of RTIs, leading to poor health outcomes. 

## 2. Methods

This is an observational case-control study involving two groups: Injured and non-injured populations.

### 2.1. Injured Population (Cases)

Cases were obtained from King Abdulaziz Medical City (KAMC) in Riyadh, the capital of SA. KAMC is equipped to treat complex injuries and has access to specialized teams including general, vascular, and orthopedic surgeons. We identified patients who were captured in the hospital’s trauma registry, which has been collecting trauma admission data since 2001. The registry captures any trauma related admission as well as those who died during or prior to hospitalization. Participants were recruited from KAMC inpatient wards following admission via the emergency department. Patients were included if they sustained non-fatal RTIs as drivers or front seat passengers and were admitted to the hospital. Patients intubated for more than two weeks were excluded due to the inability to obtain their seatbelt and mobile phone use information.

### 2.2. Non-Injured (Controls)

Four observers worked in pairs to observe vehicles at various highway overpasses and inner-city intersections in Riyadh (Figure 1). Four major highways, where significant traffic flow exists, were selected: The roads connecting Riyadh to the northern, southern, eastern, and western regions of SA. Data collection was done during weekdays in the late afternoon rush hour (between 15:00 and 18:00). For each of the four locations, 200 vehicles were observed. To record the study information and ensure safety while collecting highway data, observers stood on the sidewalks of highway overpasses. Data collection started when observers announced that they will observe the third vehicle passing their way. When selecting the vehicle, the observers specified some of the vehicle’s features (i.e., color) before documenting the driver’s seatbelt and mobile phone use.

Riyadh can be divided into nine main zones, separated by four major highways. Ten intersections were selected within each zone. Next, using computer randomization, one intersection was selected from each zone. The observers stood at a corner at each of the nine intersections where it was possible to clearly and safely inspect passing vehicles from all directions. One hundred vehicles were observed for seatbelt and mobile phone use while driving through the intersection.

To ensure accuracy of data collection, the observers stood on an elevated pedestrian overpass that allowed observing vehicles from a reasonable distance and the ability to gaze into each vehicle. For inner city intersections, the velocity of the vehicles was slow enough for the observers to record their data. All collected data were evaluated using the Kappa statistic to ensure validity and precision. There was substantial agreement between the two observers ranging from 0.70 for mobile phone use to 0.83 in seatbelt use. One observer was selected at random and used for all analyses in this study. The data used for the controls was part of a project estimating the baseline prevalence of seatbelt and mobile use of drivers in Riyadh [27]. 

### 2.3. Outcome Measures

The primary outcome measure was using a mobile phone while driving for both cases and controls. More specifically, mobile phone use was defined as usage while driving, including holding it to the ear or holding the device to text, dial, or taking photographs. Data regarding seatbelt use was also collected. 

The secondary outcome was the European Quality of Life Measure, EQ-5D [28]. This measure was used to assess disability in injured patients. EQ-5D is a self-reported health measure covering five dimensions: Mobility, pain/discomfort, self-care, usual activity, and depression/anxiety. Each dimension was measured by answering a five-level response. Disability was defined as reporting “moderate” or “severe” limitation in any of the five domains. 

The EQ-5D also includes the EuroQol visual analog scale (EQ-VAS), which requires respondents to rate their current health on a scale ranging from 0, worst possible health, to a 100, the best possible health. 

### 2.4. Data Collection

Research coordinators administered the EQ-5D in Arabic, validated in previous studies [29,30]. Two interviewers collected data from participants after obtaining informed consent. The data collected included seatbelt nonuse (driver or passenger), mobile phone use (of the driver) at the time of the accident, as well as socio-demographic and pre-injury characteristics of the participants. 

### 2.5. Sample Size

Because this was a case-control study, the sample calculation was based on the following criteria: (1) power of 90%; (2) a case-control ratio of 1:5 (five controls for each case); (3) an alpha of 5%; (4) prior prevalence of mobile phone use of 53.9% estimated by Osuagwu et al. and a confidence level of 95; and (5) a least extreme odds ratio to be detected as 1.5 [24]. The estimated required sample size was 1904 subjects: 318 cases (patients) and 1586 controls. 

### 2.6. Statistical Analysis

STATA version 15 for Mac (STATA Corp., College Station, TX, USA) was used for all statistical analyses. For cases, descriptive statistics for age, sex, mechanism of injury, transfer mode, Injury Severity Score (ISS), and other clinical or healthcare utilization variables were compared to mobile use status. Means were compared for continuous variables and proportions for categorical variables using Student’s t and Chi-square tests, respectively. To investigate the association between mobile phone use and health outcomes, the prevalence of disability in the EQ-5D domains was compared with mobile phone use status. Disability was defined as a “severe” or “moderate” problem in any of the EQ-5D domains.

For both cases and controls, the prevalence of seatbelt and mobile phone use was calculated with a 95% confidence interval. The association between mobile phone use and RTIs was determined using a simple logistic regression technique. Because the data from the cases included missing values, we performed the analysis using multiple imputation procedures to evaluate if the missing value could have influenced the findings. A *p*-value of 0.05 or lower was used as a cut-off for statistical significance. The study was reviewed and approved by the Institutional Review Board at King Abdullah International Research Center (KAIMRC).

## 3. Results

Of the 2018 participants included, 318 were cases, and 1700 were controls. Cases, overall, were relatively young (mean age 29.1 years, Table 1) and mostly male (86%). A fifth of the cases, 71 (22.5%), reported that they do not know or remember whether a mobile phone was involved in the crash. Of the remaining, 17.2% sustained RTIs while a mobile phone was used. 

The RTIs that involved a mobile phone were more severe than those not involving a mobile phone, as measured by the ISS (mean = 13.6 vs. 10.6; *p* = 0.03). This pattern was also observed for self-reported health status on the day of the interview (60.8 vs. 70.4; *p* = 0.03). Although the difference was not statistically significant, participants involved in mobile-phone related accidents were younger than participants who had accidents without mobile phone involvement (mean = 27.3 vs. 31.7; *p* = 0.12). In addition, participants involved in mobile phone related accidents were more likely to be transported by an ambulance to the hospital than their counterparts (82.9% vs. 63%; *p* < 0.05). Seatbelt use was much lower for participants involved in mobile phone-related accidents (4.8% vs. 12.2%), but the difference was not statistically significant (*p* = 0.1). 

Overall, 71.3% of the cases reported moderate or severe disability in one of the five EQ-5D domains. Patients, who reported mobile phone involvement in RTIs, were more likely to suffer from “any disability” than patients without mobile phone involvement (83.3% vs. 68.8%; *p* = 0.05, Table 2). Notably, cases involved in mobile phone-related RTIs were more likely to report they had moderate or severe disability in mobility (71.4% vs. 47.5%; *p* < 0.01), self-care (66.6% vs. 40.6%; *p* < 0.00), and usual activities (73.8% vs. 52.4%; *p* < 0.01, Table 2). Moderate to severe anxiety was also more prevalent in patients involved in accidents associated with mobile phone use, but the result was not significant (42.8% vs., 30.2, *p* = 0.06).

Seatbelt compliance was low in both groups, especially among cases with RTIs associated with mobile phone use (4.8%, Table 3). Among the controls, seatbelt compliance was 45.2% (95% CI = 41.8–48.7) for highways and 23.8% (95% CI = 21.2–26.7) for inner-city intersections. As for mobile phone use, the prevalence was higher among cases but statistically insignificant relative to controls (17.2% vs. 13.8, *p* = 0.1). For the controls, mobile phone use while driving was 15.1% (95% CI = 12.8–17.7) and 12.6% (95% CI = 10.6–15.0) at inner intersections. According to the regression analysis, using a mobile phone while driving is associated with 1.44 higher odds of sustaining a severe RTI (Table 4; *p* = 0.04).

## 4. Discussion

This study found mobile phone use while driving associated with RTIs increased the severity and prevalence of disability, and reduced HRQOL. We also found a high prevalence of seatbelt nonuse among both cases and controls. These findings highlight the need for behavioral public health interventions to reduce the impact of traffic violations on RTIs. Unless public health measures are initiated, mobile phone-related RTIs are likely to continue to have a significant effect on the population health of SA. 

Our estimate is lower than reported in literature from countries such as the United States (US), which suggests that mobile phone use while driving is associated with a fourfold increased risk of RTIs [31]. McEvoy et al. reported a similar finding in the United Kingdom (UK), where a fourfold increased risk of RTIs was reported for both hand-held and hands-free devices [16]. It is possible that patients included in our study underreported mobile phone use. Performing a sensitivity analysis, assuming that participants with missing values (22%) were involved in mobile phone-related RTIs, the estimated impact of mobile phone use on RTIs was similar to what has been reported in the US and the UK (OR = 3.5; 95% CI = 2.6–4.5). 

The prevalence of mobile phone use in the control group is also higher than in other countries. Based on different observational studies, 2.8% of drivers in Canada, 7% of drivers in the US, and 2.2% of drivers in the UK are using a mobile phone while driving [32,33,34]. However, some literature highlights a different pattern. In Australia, the prevalence of texting or browsing on phones while driving was approximately 50% for drivers [35]. In the Ukraine, 22.2% of drivers reported using their phones on a daily bases for texting or reading text messages [36]. Both studies collected self-reported habitual use data, a different approach than the method used in the current study which may explain the significant differences. 

Several previous studies investigated mobile phone use of SA drivers [22,23]. Using a self-report questionnaire in the general population, Osuagwu et al. estimated that mobile phone use was associated with a seven times higher likelihood of accidents [24]. Another study in a neighboring country, Oman, reported 90% mobile phone use while driving [37]. It is not possible to compare our findings directly to previous studies due to the differences in methodology. An essential contribution of the current study is that the association between mobile phone use and traffic accidents was not estimated, the study focused on RTIs, which has more implications for population health. 

As expected, younger individuals were more likely to incur mobile phone-related RTIs. In addition, we found mobile phone use to be associated with disability in three of the five EQ-5D domains. The likelihood that RTIs in a younger population will result in disabilities leading to a lifelong impact on population health is high. A previous study in SA with young trauma patients, mostly due to RTIs, found that one-third suffered permanent disabilities [38]. The impact of RTIs on a younger population is an essential consideration for public health officials as well as the traffic police to design prevention programs suitable for this group. Specifically as using a mobile phone while driving is considered a traffic violation in SA [39]. 

Because using a phone while driving is a traffic violation, it is fair to say that the driver was at fault in the crash. This may not always be the case. Literature does suggest, however, that mobile phone use while driving decreases response time and awareness of surroundings [40].

The use of social media applications has become increasingly popular in SA, which may underpin the behavior of driving and mobile phone usage. According to the Communications and Information Technology Commission (CITC), 91.7% of the population is using social media [41]. Literature suggests there are different levels of distractions, and different risks may be associated with various activities [33]. Additional research is required to understand exactly what injured drivers were doing while using the mobile phone (i.e., Snapchat, texting, etc.). Knowing the specific use of social media applications may guide prevention strategies to design practical approaches to promote safety, for example, partnering with social media applications to disseminate safety messages.

The present study has several implications for public health practice and policy. Firstly, the results can assist safety advocates to educate the public, policymakers, and clinicians about the dangers of using mobile phones while driving. Secondly, public health prevention programs by governmental agencies may use these results as a baseline to evaluate the effectiveness of existing and future interventions to reduce traffic violations. Traffic Police also should use this finding to guide and intensify enforcement efforts. Similar to the Healthy People 2020 program in the US, SA has launched the Saudi Vision 2030 [42]. One of the many goals of this vision is reducing the burden of RTIs. Therefore, the Saudi Vision 2030 can use our results as a baseline to monitor improved compliance in response to interventions to pave the way to improve traffic safety.

Despite existing laws, our study found a low prevalence of seatbelt use among both cases and controls (10% and 33%, respectively). Moreover, population-level estimates suggest seatbelt compliance is as low as 5% in SA though it is one of the most effective and cheapest interventions to reduce RTIs [23,43]. Estimates from studies in developed countries indicate that persons not wearing a seatbelt at the time of the injury are 5.4 times more likely to die, 1.9 times more likely to be hospitalized, and cost $268,000 more than individuals wearing seatbelts [44]. Unfortunately, seatbelt use remains a neglected issue, which may explain the low compliance rate compared to other developed countries. There is an urgent need to take a serious stand to increase compliance nationwide. 

Due to the significant burden of RTIs, the Traffic Police has implemented a few initiatives to support traffic safety. The most significant of which have been mandating seatbelt use in 2001, introducing speed cameras in 2010, and launching mobile phone and seatbelt use detection cameras in March 2018 [39]. Future studies are required to evaluate the effectiveness of this new technology in reducing seatbelt and mobile phone violations. 

Our study has several important limitations which must to be acknowledged. Firstly, because cases were obtained from a single hospital, generalizing the findings to SA is problematic. Secondly, information bias may be a threat to the validity of our study because the use of seatbelt and mobile phone in the cases were self-reported. Over 22% of the interviewed patients reported they do not know or remember whether a mobile phone was involved in the accident. It is possible that these patients chose not to report mobile phone use due to social desirability bias, fear of citation, or other legal repercussions. Thirdly, due to the nature of selecting controls, we were unable to adjust for several variables, for example age, that may act as potential confounders. In addition, the accuracy of documenting seatbelt and mobile phone use for passing vehicles, especially in highways, will likely depend on the vehicle’s speed, weather conditions, or even the observer’s ability (i.e., eyesight). Despite that, there was high agreement between observers in our study reducing the likelihood of bias. It should be noted that despite finding an association in the regression analysis, our results may underestimate the effects of mobile phone use while driving on RTIs. One reason is that our study was limited to nonfatal injuries. Nearly 9% of traffic accident victims die either before or after reaching healthcare facilities in SA [45]. We excluded patients who died because it was not possible to obtain accurate data about the actual use of a mobile phone from the deceased drivers. In addition, we excluded possible participants intubated for more than two weeks. These two exclusions may have caused an underestimation of the impact of mobile phone use on RTIs. Future studies should aim to incorporate records from phone companies to negate this limitation. Finally, the classification of mobile phone-related accidents was only done for the injured driver or passenger. The possibility exists that the other driver involved in the accident caused the accident due to being distracted by a mobile phone. Such a case was not captured in the current study and may have led to underestimating the impact of mobile phone use on RTIs. 

Nevertheless, this study has numerous strengths. Firstly, by observing drivers on Riyadh roads, we used a more representative approach to select controls which increase the likelihood to represent the underlying population. Secondly, the use of a large number of controls improved the power of the study. Thirdly, the ability to abstract data from medical records allowed examining not only the association between mobile phone use and motor vehicle accidents but also the impact on injury severity and healthcare utilization. Finally, we used a sound statistical approach to manage missing values taking uncertainty into account. 

## 5. Conclusions

In summary, we found mobile phone use while driving to be associated with RTIs, an increased severity and higher prevalence of disability, and a reduction in the HRQOL. This result emphasizes an alarming concern, namely that using a mobile phone while driving is a significant threat to traffic safety and population health in SA. Seatbelt use remains extremely low compared to developed nations and requires significant improvement. Prevention programs may use these results to educate the public, advocate for increased enforcement, and to raise investment in public health programs to reduce RTIs and improve the Saudi Arabian population’s health. 

## Figures and Tables

**Figure 1 ijerph-16-02706-f001:**
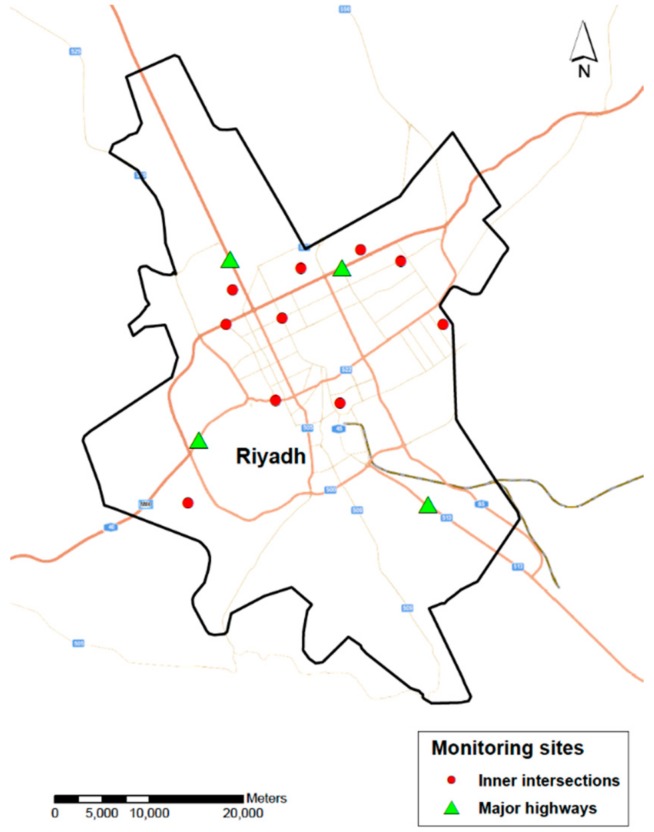
Observation sites for seatbelt and mobile violations in Riyadh.

**Table 1 ijerph-16-02706-t001:** Descriptive characteristics of cases admitted at King Abdulaziz Medical City, 2017–2018.

Variable	Mobile Phone Used *N* = 42	No Mobile Phone Used *N* = 202	Total Population *N* = 247	*p*-Value
Mean age	27.37	31.72	29.34	0.12 ^
Male %	90.48	85.15	86.07	0.82 *
Mechanism of injury %				
Occupant	26.19	35.64	34.02	0.23 *
Driver	73.81	64.36	65.98	
Transport to hospital %				
EMS	82.93	63.02	66.52	0.01 *
Private vehicle	17.07	36.98	33.48	
Pre-injury reported health mean (SD)	99.64 (1.71)	97.69 (7.36)	98.25 (6.18)	0.09 ^
Post injury Reported Health mean (SD)	60.79 (26.79)	70.44 (25.25)	69.04 (25.44)	0.03 ^
GCS mean (SD)	13.54 (3.24)	14.04 (2.50)	13.55 (3.05)	0.27 ^
ISS mean (SD)	13.64 (9.00)	10.60 (7.55)	11.36 (14.81)	0.03 ^
Length of stay mean (SD)	17.29 (27.16)	16.66 (27.71)	20.68 (34.77)	0.89 ^
ICU admission %	23.81	27.23	26.64	0.21 *
Surgery %	9.52	6.93	7.38	0.56 *
Head injury %	26.83	15.10	17.17	0.07 *
Trauma team activation %	28.57	19.80	21.31	0.20 *
Seatbelt use %	4.88	12.24	10.97	0.17 *

* Chi-square test; ^ Student’s t test; ISS, Injury Severity Score (higher indicates worst injuries); GCS, Glasgow Coma Scale (lower indicates worst injuries); ICU, Intensive care unit.

**Table 2 ijerph-16-02706-t002:** Frequencies of the five European Quality of Life (EQ-5D) dimensions by mobile phone use while driving.

	Mobile Use	No Mobile Use	*p*-Value
Mobility			
No problem (%)	12 (28.57)	106 (52.48)	0.01
Moderate/Severe (%)	30 (71.43)	96 (47.52)	
Self-care			
No problem (%)	14 (33.33)	120 (59.41)	<0.01
Moderate/Severe (%)	28 (66.67)	82 (40.59)	
Usual activities			
No problem (%)	11 (26.19)	96 (47.52)	0.01
Moderate/Severe (%)	31 (73.81)	106 (52.48)	
Pain/discomfort			
No problem (%)	22 (52.38)	113 (55.94)	0.67
Moderate/Severe (%)	20 (47.62)	89(44.06)	
Anxiety			
No problem (%)	24 (57.14)	141 (69.80)	0.11
Moderate/Severe (%)	18 (42.86)	61 (30.20)	
Any disability			
No problem (%)	7 (16.67)	63 (31.19)	0.06
Moderate/Severe (%)	35 (83.33)	139 (68.81)	

**Table 3 ijerph-16-02706-t003:** Seatbelt use among cases and controls by mobile use status.

Group	Prevalence of Seatbelt Use % (95% CI)
**Cases (overall)**	9.0 (6.2–12.9)
Mobile phone involvement	4.8 (1.1–18.4)
No mobile phone involvement	12.2 (8.3–17.6)
**Controls (overall)**	33.9 (31.7–36.2)
Mobile phone involvement	15.3 (11.2–20.5)
No mobile phone involvement	36.9 (34.4–39.4)

**Table 4 ijerph-16-02706-t004:** Logistic regression analyses of the association between mobile use and traffic injuries.

Variables	Logistic Regression of Odds of Traffic Injuries Imputing Missing Values OR (95% CI)	*p*-Value
Mobile use		
Controls	Reference	
Cases	1.44 (1.00–2.05)	0.04
*N*	2018

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
