# Peer review of "The Association between Mobile Phone Use and Severe Traffic Injuries: A Case-Control Study from Saudi Arabia"

_ijerph, 2019, doi:10.3390/ijerph16152706_

Reviewer 1 Report

I recommend the authors submit to a journal specialising in road traffic as there is little environmental content and thus it is not a best match for this readership.

Formatting and grammar issues throughout, avoid short, 2 sentence paragraphs; limitations stated twice in the discussion.

Unclear why the authors want to study what type of social media applications were being used while driving and what effect that will have on prevention.

Analysis of literature on mobile phone use in the region lacking.

Author Response

Reviewer # 1

recommend the authors submit to a journal specialising in road traffic as there is little environmental content and thus it is not a best match for this readership.

We value the reviewer’s comment, but this manuscript was submitted to the special issue about road injury. The section of this special issue is “Health Behavior, Chronic Disease, and Health Promotion." Thus, it is our belief that it is the right place for it since a mobile phone while driving is a major emerging threat to traffic safety worldwide. This paper focuses on drivers' behavior, which inevitably impacts driving environments. Being in this issue would increase the readability of this paper and facilitate public health impact.  

Formatting and grammar issues throughout, avoid short, 2 sentence paragraphs; limitations stated twice in the discussion.

The manuscript was revised and edited extensively. Limitations were not mentioned twice, but the first part was to outline the potential explanation of the findings. In response, we merged that with the limitations part toward the end of the manuscript.

Unclear why the authors want to study what type of social media applications were being used while driving and what effect that will have on prevention.

This may facilitate future public health interventions, such as partnering with a particular social media app (i.e., Snapchat) to disseminate tailored messages to users of these apps. In response, we highlighted this further on lines 237-240.

Analysis of literature on mobile phone use in the region lacking.

Few studies have explored the topic in our region. This another reason we feel this paper is a good addition to the literature. In response, we added a study about a neighboring country of Oman (lines 214-215). 

Reviewer 2 Report

This is an interesting paper that investigates the effect of mobile phone use on injury/health outcome. The case and control design seems proper and the results are in line with literature.  The following issues need to be addressed before the reviewer has a favourable recommendation:

Major

1.      Section 2:1: the subjects were recruited from the participating hospital. The authors should explicitly state this in the manuscript title and abstract.

2.      Following the comment #1, please clarify how the patients were identified to have involved in a crash. By e-code? Or any other method? If by e codes, were they reliable? In general, e-codes were not complete in rural hospitals; and emergency physicians in general do not record E codes completely.

3.      Following the comment #1, patients intubated for more than two weeks were excluded. Also, if reviewer understands the paper fully, outpatients that were involved in crashes were not enrolled in the study. Please address how this sample selection (those who were hospitalised for more than 2 weeks and those who were outpatients) would affect your results. If such an issue is not addressed properly or statistically, the results will be biased.

4.      Observing phone use and seat belt use from the passing cars would be dependent on several factors such as vehicle’s speed and/or how transparent car window is. Past studies have adopted a similar way to collect, for example, helmet use data. Although the inter-rater reliability test seems high, the reviewer finds it difficult to accurately observe helmet use or phone use from passing vehicles.

Minor

1.      Please state whether this study was approved an IRB.

2.      Several well-known studies in this area should be cited and discussed, e.g., DA Redelmeier, RJ Tibshirani (1997) from NEJM, and other studies from AAP.

Author Response

Reviewer # 2

This is an interesting paper that investigates the effect of mobile phone use on injury/health outcome. The case and control design seems proper and the results are in line with literature.  The following issues need to be addressed before the reviewer has a favourable recommendation:

Major

1.     Section 2:1: the subjects were recruited from the participating hospital. The authors should explicitly state this in the manuscript title and abstract.

We revised the abstract to address the reviewer’s comment. We do not feel the title should be changed as only cases were ascertained from the hospital while controls where not.

2.      Following the comment #1, please clarify how the patients were identified to have involved in a crash. By e-code? Or any other method? If by e codes, were they reliable? In general, e-codes were not complete in rural hospitals; and emergency physicians in general do not record E codes completely.

Patients were identified from the hospital’s trauma registry. E codes are still not implemented in this hospital. Those who sustain a motor vehicle injury as occupants or drivers are grouped into one group. Identifying those patients is not very challenging since only a few patients are admitted to the hospital in a given week. In response, further details were included on lines 96-99.  

3.     Following the comment #1, patients intubated for more than two weeks were excluded. Also, if reviewer understands the paper fully, outpatients that were involved in crashes were not enrolled in the study. Please address how this sample selection (those who were hospitalised for more than 2 weeks and those who were outpatients) would affect your results. If such an issue is not addressed properly or statistically, the results will be biased.

We acknowledge the reviewer’s concern. However, outpatient visits are not included among our study population since our inclusion criteria were focused on severe injuries defined by admission to the hospital. Excluding intubated patients will likely underestimate the effect of using a mobile phone on the risk of sustaining severe road traffic injuries. Nevertheless, our decision aimed to ensure the validity of the recall period as those intubated for a long time may not remember the mobile phone status accurately. In response, we highlighted this on page 6, line 277 of the limitations.

4.     Observing phone use and seat belt use from the passing cars would be dependent on several factors such as vehicle’s speed and/or how transparent car window is. Past studies have adopted a similar way to collect, for example, helmet use data. Although the inter-rater reliability test seems high, the reviewer finds it difficult to accurately observe helmet use or phone use from passing vehicles.

The investigators realized this potential challenge early before data collection. Therefore, they ensured when data collectors are positioned to document mobile phone and seatbelt use, that: 1) an observation site is safe 2) they can clearly document drivers' seatbelt mobile phone use. In addition, the timing of observing vehicle was in the middle of daylight, which makes observing passing vehicles clearer. As for highway passing vehicles, the observers stood on an elevated pedestrian overpass that gave an advantage of observing vehicles from a reasonable distance, coupled with the ability to gaze through the inside of each vehicle. As for data collected in inner intersections, the velocity of the vehicles was slow enough for the observers to record their data. In response, we highlighted this further on lines 119-122.

Minor

1.     Please state whether this study was approved an IRB.

We confirm that we had already received an IRB approval before conducting this study. In response, this was highlighted on line 157.

2.      Several well-known studies in this area should be cited and discussed, e.g., DA Redelmeier, RJ Tibshirani (1997) from NEJM, and other studies from AAP.

In response, we added this study as well as additional references. 

Reviewer 3 Report

This is a very nice piece of research and deserves to be published, However I have some remarks and suggestions and I trust that the authors will have a look and use it for further improvement of this manuscript.

I have a couple of general and more detailed comments.

General

This ms has a very strong public health lense and focus. That is not a surprise if I see the affiliations of the authors. However: traffic and traffic injuries cannot be approached only by the public health perspective. RTIs have certainly a public health component, but can also be approached from a transport, enforcement, human factors, public policy in general, etc. perspective. I invite the authors to read their ms carefully and see if other perspectives can be used to make the story more balanced.

I observe a strong focus on Saudi Arabia and USA in the references. I am not expecting that from a scientific paper. I recommend to look at references from other parts of the world, especially Europe and Australia.

The design of the study is sound. The authors are brave enough to present the inherent weaknesses of their approach and that is appreciated.

Specific

line 22: ...."the third leading cause of death". This is rather high. It would be good to compare this with other high-motorised countries. The mortality rate is extremely high: WHO estimates 28.8. It makes sense to include this in your ms and offer an international perspective on that.

line 39: you use "almost". Can you mention countries in the world that don't asses their road traffic toll as a significant threat to public heatlh?

line 45 and source [4]. Please inform the reader about the quality of police statistics in SA. As you may know many countries suffer from the fact of so-called underreporting, especially in case of injuries.

line 55 etc. I am not satitsfied by the way the authors introduce distracted driving and use of mobile phones as a road safety problem. This requires more introduction and I suggest to tap from the Human Factors literature on that: slower reactions, less control, less attention to relevant visual information, increase of mental workload, etc. If using mobile phones will not be compensated by some form of behaviour adapatation, risks will increase. Go for example to a recent thesis that I found "The effects of using mobile phones and navigation systems during driving" (2018). This is a PhD-study of Allert Knapper.

line 74. It is not fully clear what this HRQOL-perspective is really adding to this research.Please, be more specific what you mean with survivors: are these road traffic injuries? Are these families, communities suffering from road fatalitries or road injuries? If so, why only include consequences of helath and not economic consequences for example.

line 86. the authors speak about association. That is correct, given the design of the study. But, the ms will be stronger if association could be replaced by causal relationship. So, in case you find an odds ratio >1, how to interpret that in your study?

line 92 and further: please explain how you measure prevalence of using a mobile-phone resp. seat belt wearing. Did you use any information given by police or bystanders? Furthermore, what is your 'culture' telling you about how honest people are on admitting non-use, or of illegal activities?

line 101: It might be nice to add a map of Riyadh and show the locations of your measurements.

line 114: we have to distinguish between handsfree and handheld. Can you explain how observers could detect handsfree mobile use, if they are positioned outside the car? Are we only talking about drivers (or also other car occupants)? 

line 138: the description of the way you arrive at your sample size is too short. Make this a more serious description in which you 'explain' why you picked these criteria.

line 144.the authors do not present a methodology. The title has case-control in it, but no reference is being made in the text to that. Now it is not obvious why to include variables as listed here. I recommend to add a paragraph on the methodology. Then it is of relevance to learn how you to correct for differences between cases and controles on relevant variables (age, for example). And I want to learn why seat belt wearing has been included in this study

line 161: I read males (86%). Please, correct if I am wrong: I thought that females were not allowed to drive a car (recently chnaged?).

line 170. I am shocked by the dramatic low percentage of seat belt use in SA. This is the lowest percentage I have ever seen in a study or in my work in dozens of countries all over the world. I invite the authors to comment on this, using words which reflect the seriousness of this problem.

line 190. You arrive at an odds ratio of 1.44 for use of mobile phones and page 6 presents reason why this 1.44 is most probably too low. Will it be possible to correct quatitatvely for these flaws, as you did for 'information bias'?

line ?? please, give prevalences of mobile phone use for different locations (for the controls) in your ms, just to learn about the variation over locations.

line 243: please refrain from using causing in this study. That is a very critical issue in safety research (in general).

line 276 source 42: I haven't read that document, but I cannot believe its results. Please, explain in more detail what happened here and assess if this is a sound/scientific study

line 289 and further: to conclude: I found it more alarming to read about the extremely low wearing rates of seat belts, than on the prevalences/risks of using mobile phones in SA. Your recommendations at the end are somewhat broad, mild and general, and not very inspiring. Please give that your thoughts and try to be more specific.

Author Response

Reviewer # 3

This is a very nice piece of research and deserves to be published, However I have some remarks and suggestions and I trust that the authors will have a look and use it for further improvement of this manuscript.

We thank the reviewer for the positive comment. 

General

This ms has a very strong public health lense and focus. That is not a surprise if I see the affiliations of the authors. However: traffic and traffic injuries cannot be approached only by the public health perspective. RTIs have certainly a public health component, but can also be approached from a transport, enforcement, human factors, public policy in general, etc. perspective. I invite the authors to read their ms carefully and see if other perspectives can be used to make the story more balanced.

The focus of the International Journal of Environmental and Public Health is on public health, which reflects the interest of its readers, and this is the main reason we focus on this component. In addition, many of the public health interventions overlap with the factors highlighted by the reviewer, such as human factors and public policy. In response, we revised the manuscript and highlighted the role of other perspectives, especially enforcement (line 251, 299).

I observe a strong focus on Saudi Arabia and USA in the references. I am not expecting that from a scientific paper. I recommend to look at references from other parts of the world, especially Europe and Australia.

We acknowledge the reviewer’s point and in response, included discussion several studies from Australia and other countries (line 219-222).

The design of the study is sound. The authors are brave enough to present the inherent weaknesses of their approach and that is appreciated.

We appreciate this feedback. 

Specific

line 22: ...."the third leading cause of death". This is rather high. It would be good to compare this with other high-motorised countries. The mortality rate is extremely high: WHO estimates 28.8. It makes sense to include this in your ms and offer an international perspective on that.

In response, we added this on lines 44-46. 

line 39: you use "almost". Can you mention countries in the world that don't asses their road traffic toll as a significant threat to public heatlh?

We see the reviewer’s point. The use of “almost” here was to reflect how can the burden significantly differ from one country to another (i.e., Sweden: 2.8 deaths for 100,000 vs. the Dominican Republic: 41.7 per 100,000). In response, we removed this word. 

line 45 and source [4]. Please inform the reader about the quality of police statistics in SA. As you may know many countries suffer from the fact of so-called underreporting, especially in case of injuries. 

As reported in the WHO report of 2013, Saudi Arabia is classified among “Countries without eligible death registration data." This is in part due to the limitations in police statistics. One study compared police reported deaths and hospital deaths and found around 31% underreporting of traffic deaths. In response, we included this in the revised manuscript (line 79-82). 

line 55 etc. I am not satitsfied by the way the authors introduce distracted driving and use of mobile phones as a road safety problem. This requires more introduction and I suggest to tap from the Human Factors literature on that: slower reactions, less control, less attention to relevant visual information, increase of mental workload, etc. If using mobile phones will not be compensated by some form of behaviour adapatation, risks will increase. Go for example to a recent thesis that I found "The effects of using mobile phones and navigation systems during driving" (2018). This is a PhD-study of Allert Knapper.

In response, we expanded the introduction further and incorporated literature suggested by the reviewer (lines 56-62). 

line 74. It is not fully clear what this HRQOL-perspective is really adding to this research. Please, be more specific what you mean with survivors: are these road traffic injuries? Are these families, communities suffering from road fatalitries or road injuries? If so, why only include consequences of helath and not economic consequences for example.

The reason for including this is to evaluate the role of mobile phone use not only in sustaining injuries but also with the magnitude of disability that occurs as a result of injuries. As for the use of the word "survivor," it refers to those who suffered the injuries and did not die as a result. In response, this was clarified further (lines 79-80). The economic consequences are another aspect that captures the burden aside from death and is an excellent idea for future research, but data availability of hospital cost is going to be a significant challenge.  

line 86. the authors speak about association. That is correct, given the design of the study. But, the ms will be stronger if association could be replaced by causal relationship. So, in case you find an odds ratio >1, how to interpret that in your study?

We agree that a causal relationship is stronger, but causation is a strong claim in epidemiology that we cannot assume with the current design. As usually done in clinical and epidemiologic papers, odds ratios are interpreted as reflecting the increased likelihood of the outcome in response to a given exposure. 

line 92 and further: please explain how you measure prevalence of using a mobile-phone resp. seat belt wearing. Did you use any information given by police or bystanders? Furthermore, what is your 'culture' telling you about how honest people are on admitting non-use, or of illegal activities?

None of the data collected was based on bystanders or the police, but rather by trained observers. As highlighted in response to an earlier comment, police data has many limitations.  Because seatbelt nonuse and mobile phone use are both considered traffic violations, self-reported use of mobile phone and seatbelts will likely to be biased due to fear of citations or social desirability bias. This has been suggested in other countries as well (Türker Özkan, Prasanthi Puvanachandra, Timo Lajunen, Connie Hoe, Adnan Hyder, 2012). 

line 101: It might be nice to add a map of Riyadh and show the locations of your measurements.

In response, we added the map where observations had taken place (Figure1) on page 4. 

line 114: we have to distinguish between handsfree and handheld. Can you explain how observers could detect handsfree mobile use, if they are positioned outside the car? Are we only talking about drivers (or also other car occupants)?  

The data recorded in this study focused on hand-held phone use by the driver. Handsfree use of mobile phones (i. Bluetooth or car speakers) was not considered since it is traffic violation in Saudi Arabia as well as most countries. Other car occupants use of mobile phones was not recorded. In response, this was clarified and specified further (lines 135-137).

line 138: the description of the way you arrive at your sample size is too short. Make this a more serious description in which you 'explain' why you picked these criteria.

The process used in the paper is the standard in sample calculation for case-control studies. Most of the criteria selected are what is commonly practiced in epidemiologic papers. For example, an alpha of 0.05 and power of 90%. We selected an odd ratio of 1.5 as a meaningful difference to be detected and had we selected a higher estimate; our sample would have been smaller. Therefore, we aimed to be conservative to ensure capturing meaningful differences. In response, we highlighted this further on 139-143. 

line 144.the authors do not present a methodology. The title has case-control in it, but no reference is being made in the text to that. Now it is not obvious why to include variables as listed here. I recommend to add a paragraph on the methodology. Then it is of relevance to learn how you to correct for differences between cases and controles on relevant variables (age, for example). And I want to learn why seat belt wearing has been included in this study

The first line of the method was stating the study methodology. Seatbelt use was included due to its importance in traffic safety. As the reviewer pointed out in later comments, the prevalence in Saudi Arabia is shockingly low. 

line 161: I read males (86%). Please, correct if I am wrong: I thought that females were not allowed to drive a car (recently chnaged?).

Yes, it is correct. This includes females front passengers in the vehicle. This was highlighted further in the revised manuscript (line 104-105).   

line 170. I am shocked by the dramatic low percentage of seat belt use in SA. This is the lowest percentage I have ever seen in a study or in my work in dozens of countries all over the world. I invite the authors to comment on this, using words which reflect the seriousness of this problem.

We could not have agreed more. Seatbelt use is one of the cheapest and cost-effective tools to facilitate traffic safety. Yet, it remains a neglected priority, and it is why studies like this can help shed more light and hopefully engage policymakers to invest in increasing compliance. In response, we highlighted this further in the revised manuscript (lines 274-276; 312-313).

line 190. You arrive at an odds ratio of 1.44 for use of mobile phones and page 6 presents reason why this 1.44 is most probably too low. Will it be possible to correct quatitatvely for these flaws, as you did for 'information bias'?

These are issues inherited from the was data was ascertained; in this case, the use of self-report to identify mobile use. Aside from what we did in reclassifying those who stated they do not remember whether mobile use was involved, we are unaware of other methods to address underreporting of mobile use.

line ?? please, give prevalences of mobile phone use for different locations (for the controls) in your ms, just to learn about the variation over locations.

In response, this was added to the manuscript on line 188-192.

line 243: please refrain from using causing in this study. That is a very critical issue in safety research (in general).

We agree, and the manuscript was revised accordingly.

line 276 source 42: I haven't read that document, but I cannot believe its results. Please, explain in more detail what happened here and assess if this is a sound/scientific study

This U.S. study evaluated whether states the ban of a hand-held phone, while driving was associated with lower road-side, observed hand-held mobile phone usage. To asses this association, they linked roadside observations of drivers with whether or not a state had a universal ban on mobile phone use while driving. In our assessment, this was a sound study using a nationally representative survey of road users. Trained observers collected information on the selected intersections about mobile phone use while driving via direct observation. However, several potential confounders may explain differences between states other than the presence of the law. For example, sites used to collect the observations may differ between states with or without a mobile phone ban in a way that leads to different estimates due to factors such as traffic congestions. In addition, they did not have data on some states after the mobile phone ban, which may lead to some bias. In response, the statement about this study was revised to reflects the uncertainty of the results (lines 281-283).  

line 289 and further: to conclude: I found it more alarming to read about the extremely low wearing rates of seat belts, than on the prevalences/risks of using mobile phones in SA. Your recommendations at the end are somewhat broad, mild and general, and not very inspiring. Please give that your thoughts and try to be more specific

We could not have agreed more as stated to an earlier comment. We believe there is an urgent need to take a serious stance to increase compliance levels. In response, specific mention was made to increase enforcement of this violation as a means to increase compliance (lines 313-316).  

Round  2

Reviewer 1 Report

Limitations should be in one paragraph.

Author Response

In response, the paper was edited by an English native speaker and the limitations section was put in one paragraph. 

Reviewer 2 Report

The reviewer still finds it difficult to be convinced by authors' statements regarding observing phone use and seat belt use from the passing cars.  This is not an unique approach to collect data; however, in practice, prosecuting a driver using phone or not wearing belt through photos is controversial.  

Accuracy of the data would depends on a multiple factors, for example, whether the windows were transparent enough, vehicle speeds, weather conditions, and even the observers themselves (eyesight). Without stopping the vehicles, the data were likely to be inaccurate. 

Author Response

The reviewer still finds it difficult to be convinced by authors' statements regarding observing phone use and seat belt use from the passing cars.  This is not an unique approach to collect data; however, in practice, prosecuting a driver using phone or not wearing belt through photos is controversial.   Accuracy of the data would depends on a multiple factors, for example, whether the windows were transparent enough, vehicle speeds, weather conditions, and even the observers themselves (eyesight). Without stopping the vehicles, the data were likely to be inaccurate. 

We agree that the accuracy of the data can depend on multiple factors. For data collection that was conducted at intersections, our approach was actually similar to what has been conducted in other countries such as in the National Occupant Protection Use Survey (NOPUS) where trained observers document seatbelt or mobile use at intersections. Because there were in intersections, passing cars were slow enough to allow for clear visibility. We should acknowledge that no strategy is perfect, but highlighting any potential limitation is instrumental in case they exist. To capture seatbelt and mobile use of drivers in highways, stopping them to do so will also result in inaccurate data and will underestimate the true prevalence of violation. As for tinted windows, these are also likely to underestimate the prevalence of traffic violations of not wearing seatbelt use and using a phone while driving because tinted windows are considered traffic violation in Saudi Arabia. The weather in Riyadh is mostly hot and dry in over eight months of the entire year. In response to the reviewer’s concern, we highlighted this further (lines 1063-1066).

Reviewer 3 Report

I thank the authors for their responses to my earlier reamrks, questions and suggestions. They did a good job on that.

Only a couple of relatively minor issues (and one major:#2) are left and I ask the authors to pay attention to them, and I trust that this will result in some amendments of the draft ms.

Abstract. The authors agree with me that not wearing a seat belt is a major issue in SA. However, this is not reflected in the Abstract. I recommend to add one sentence to that.

An oddsratio of 1.44 for using a mobile while driving is considerably lower than reported in the literature (OR up to 3). This deserves more attention of the authors. Or they can give good reasons for that, or they argue that flaws in this research (e.g. self-reported behaviour) is a potential explanation for that. At least this topic deserves serious attention.

line 84-85. This sentence is not ok, so please reformulate this one. The authors can refer to the international literature (IRTAD) and/or to the WHO Global status report (2016 fatality data for SA: 9031 vs. 9311).

line 243. It is not clear why this sentence of 'implicit assumption...' is of relevance in this study. It might be better to remove this sentence, or explain exactly what the authors intend to say.

line 259. The formulation of this sentence is very soft (police can use ...). Perhaps the authors have good reasons to do this. An explanation might be that police enforcement is (almost) absent in SA and drivers know that! How to deal with this subject? And is you assume that more/better police enforcement is not feasible, what to do next?

line 276 Reference 49. I have serious doubts about this research (as I said before), simply because I don't trust the results. On my earlier remarks on this the authors agree, but I don't read that in their second version (only in the response to me)

line 295 I was expecting as a definition for 'phone-related crashes 'that at least one of involved drivers was using a mobile phone. But I understand that is not the case. Only this is known of the injured driver. Is that correct? In that case, an underestimation is certain, in stead of ...may lead .... And it makes sense to consider another definition of phone-related crashes. Furthermore, why is it helpful to make a distinction between drivers and passengers in this sentence?

Author Response

I thank the authors for their responses to my earlier reamrks, questions and suggestions. They did a good job on that.

Thank you, we truly appreciate all the help to improve the quality of the manuscript. 

Only a couple of relatively minor issues (and one major:#2) are left and I ask the authors to pay attention to them, and I trust that this will result in some amendments of the draft ms.

Abstract. The authors agree with me that not wearing a seat belt is a major issue in SA. However, this is not reflected in the Abstract. I recommend to add one sentence to that.

In response, we added “The low prevalence of seatbelt use is alarming and requires significant improvement.” to the abstract (lines 33-34). 

An odds ratio of 1.44 for using a mobile while driving is considerably lower than reported in the literature (OR up to 3). This deserves more attention of the authors. Or they can give good reasons for that, or they argue that flaws in this research (e.g. self-reported behaviour) is a potential explanation for that. At least this topic deserves serious attention.

We did acknowledge that previously (lines 700-701). In response, we highlighted the issue of self-reported answers and its potential role in underestimating the association (697-699). Another issue we highlighted that may play a role was the exclusion of deaths and those intubated for more than two weeks (lines 1069-1073). 

line 84-85. This sentence is not ok, so please reformulate this one. The authors can refer to the international literature (IRTAD) and/or to the WHO Global status report (2016 fatality data for SA: 9031 vs. 9311).

In response, the sentence was reformulated to “A study reported that police underestimate annual deaths due to traffic accidents.”As for using the WHO report, we included that on line 93-94 “According to a report by the World Health Organization, SA has a traffic mortality rate of 24.8 per 100,000, which is substantially higher than countries (i.e., Qatar: 14 per 100,000) with a similar demographic and driving environment.”

line 243. It is not clear why this sentence of 'implicit assumption...' is of relevance in this study. It might be better to remove this sentence, or explain exactly what the authors intend to say.

We acknowledge the reviewer’s concern. In response, the sentence was rephrased to “Because using a phone while driving is a traffic violation, it is fair to say that the driver was at fault in the crash.” (lines 725-726).

line 259. The formulation of this sentence is very soft (police can use ...). Perhaps the authors have good reasons to do this. An explanation might be that police enforcement is (almost) absent in SA and drivers know that! How to deal with this subject? And is you assume that more/better police enforcement is not feasible, what to do next?

On the contrary, we firmly believe that improved enforcement is feasible and more research like ours will help facilitate that. In response, we changed the sentence to “Traffic Police also should use this finding to guide and intensify enforcement efforts.” (lines 741-742).

line 276 Reference 49. I have serious doubts about this research (as I said before), simply because I don't trust the results. On my earlier remarks on this the authors agree, but I don't read that in their second version (only in the response to me)

We did change the sentence in the revised manuscript to highlight the uncertainty of the estimate provided by that paper (from stating the effect was a 50% decline to a range between 38% and 63%). In response to the reviewer’s continued concern, we removed the sentence and the reference in the revised manuscript. 

line 295 I was expecting as a definition for 'phone-related crashes 'that at least one of involved drivers was using a mobile phone. But I understand that is not the case. Only this is known of the injured driver. Is that correct? In that case, an underestimation is certain, in stead of ...may lead .... And it makes sense to consider another definition of phone-related crashes. Furthermore, why is it helpful to make a distinction between drivers and passengers in this sentence?

Yes, correct. In response, the definition was changed to “the classification of mobile phone-related accidents was only done for the injured driver. The possibility exists that the other driver involved in the accident caused the accident due to being distracted by a mobile phone. Such a case was not captured in the current study and may have led to underestimating the impact of mobile phone use on RTIs.” (lines 1074-1077). 

Round  3

Reviewer 2 Report

The authors have adequately addressed my previous comments and suggestions. I am pleased to recommend publication of the paper.  One final note is that the authors may consider explicitly indicating their research limitations regarding data accuracy which were surely depending on the vehicle speeds, weather conditions, and/or even the observers themselves (eyesight). 

Author Response

We thank the reviewer for the positive response. To address the reviewer’s latest comment, we added that “In addition, the accuracy of documenting seatbelt and mobile phone use for passing vehicles, especially in highways, will likely depend on the vehicle’s speed, weather conditions, or even the observer’s ability (i.e. eyesight).” (line 285-288).